# Time to doubling of serum creatinine in patients with diabetes in Ethiopian University Hospital: Retrospective follow-up study

**Adeladlew Kassie Netere** [ID]*, **Ashenafi Kibret Sendekie** [ID]

Department of Clinical Pharmacy, School of Pharmacy, College of Medicine and Health Sciences, University of Gondar, Gondar, Ethiopia

* kassieadeladlew21@gmail.com

## Abstract

**Data Availability Statement:** All data generated or analysed during this study is included in this published article and is available as the supplementary information files.

### Background

Diabetic kidney disease is one of the long-term microvascular complications of diabetes. Doubling of serum creatinine is an important biomarker and predictor of diabetic kidney disease for patients with diabetes. This study aimed to determine the time in which the serum creatinine level is doubled measured from the baseline in patients with diabetes in Ethiopian University Hospital.

### Methods

Analysis of the patients with diabetes medical records was employed retrospectively for five years from 2016 to 2020 in the University of Gondar Comprehensive Specialized Hospital. The Kaplan-Meier procedure was used to predict the time to which the serum creatinine level was doubled measured from the baseline value, while the Log-rank test and cox-proportional hazard regression models were employed to show significant serum creatinine (SCr) changes against the predictor variables.

### Results

Among the total of 387 patients with diabetes, 54.5% were females with a mean age of 61.1 ±10.3 years. After 5-years of retrospective follow-up, 10.3% of patients with diabetes had doubled their serum creatinine level computed from the baseline values. The baseline and last SCr levels (measured in mg/dL) were 0.87 (±0.23) and 1.0(±0.37), respectively. This resulted in a mean SCr difference of 0.12±0.38 mg/dL. The SCr score was continuously increasing uninterruptedly for five years and measured as 0.94, 0.95, 0.94, 1 and 1.03 mg/dL, respectively. The average survival time taken for the serum creatinine to be doubled computed from baseline was 55.4 months (4.6 years). Patients treated with greater than or equal to 30 IU NPH were found 3.3 times more likely to have higher risks of doubling the serum creatinine level (DSC); with HR of 3.29 [(95%CI); 1.28–8.44: P = 0.013].

**Funding:** We did not get any outside financial support to conduct this study.

**Competing interests:** All the authors declared that they have no conflict of interest and contributed equally to this work.

**Abbreviations:** CKD, chronic kidney disease; DBP, diastolic blood pressure; DKD, diabetic kidney disease; DN, diabetic nephropathy; DSC, doubling of serum creatine; eGFR, estimated glomerular filtration rate; ESKD, end stage kidney disease; FBS, fasting blood sugar; HTN, hypertension; NPH, neutral protamine Hagedorn; SBP, systolic blood pressure.

## Conclusion

Compared with the baseline level, a significant proportion of patients with diabetes were found to have doubling of serum creatinine DSC within less than five years around four and half years. A continuous increasing in the SCr level was noted when measured from the baseline scores. Therefore, to preserve the renal function of patients with diabetes, close SCr level monitoring and regular follow-up would be recommended in combined with effective therapeutic interventions.

## Introduction

Diabetes mellitus (DM) is alarmingly increasing in underdeveloped countries and results in diabetic kidney disease (DKD) [1, 2]. The increase in this microvascular complication is accompanied with raising in DM prevalence, which requires cautious prevention mechanisms to delay or minimize chronic complications associated with the disease. According to the World Health Organization (WHO) reports, diabetic mellitus is rapidly surging throughout the globe, and major cause of blindness, kidney failure, heart attacks, stroke and lower limb amputation, and it caused for 1.5 million diabetes-related deaths in 2019 [3]. DM is also increasing in the sub-Saharan countries, and more than two and half million adults have been living with diabetes in Ethiopia [4]. This makes Ethiopia as one of the populous countries living diabetes in the Sub-Saharan Africa region, where the prevalence has been dramatically increased from 3.8% to 5.2% [5].

Diabetes mellitus causes different macrovascular and microvascular complications, and about forty percent of patients with DM develop DKD [6, 7], which further progresses to chronic kidney diseases (CKD) [6], end stage kidney disease (ESKD), morbidity, mortality, and increased treatment costs [8, 9]. Further, CKD is topmost health treats for about 8–12% of the adult population groups in industrialized nations [10, 11], and generally remains the global public health perils [12]. Moreover, CKD is linked to as utmost cause of increase in ESKD, all-cause mortality and cardiovascular (CV) diseases even in its early stages [13, 14].

In real clinical situation, performing studies on the progression of diabetes toward the development of DKD development is challenging and time consuming [15]. Furthermore, because of the DKD is gradually progressing to chronic renal complications and higher drop-out rates have been recorded due to the CV events, practical studies become challenging. Thus, such study trials require larger participants and a longer time for follow-up. Therefore, to solve these challenges, researchers have usually used a composite end point of death, ESKD and doubling of serum creatinine (DSC) [16, 17]. The DSC can be simple to monitor and is almost comparable to a halving of the estimated glomerular filtration rates (eGFR), and it accounts for the substances of the events containing the composite renal end points [16, 17]. The components of the DSC are well-recognized and accepted parts of the composites, because of the changes observed in the serum creatinine (SCr) through longer-time, it is implicate to reflect the structural kidney function deteriorations [18–20]. Generally, DSC has been broadly applied and acknowledged in DKD progression study trials [21–23].

Since DSC is a significant biomarker and prognosticator of DKD for patients with diabetes and is substantially related to the consequential hazards of clinical end points such as ESKD and death, it has been ever more accepted as a surrogate end points in CKD development studies [24, 25]. Because the DSC is an intermediate incidence that happened earlier the ESKD is

concurred using it as surrogates end points have numerous advantages. In addition, it might theoretically shorten the follow-up times and having included the composite end points by considering about half to sixty percent of the renal events [26, 27]. It can also be simply used in time-depended or survival analysis since it is a dual outcome and approximately agrees to halving of GFR [16]. Nevertheless, the use of DSC as surrogate end point is not without drawbacks and controversies. Therefore, it is important to use it by considering its shortcomings, such as SCr is an imprecise biomarkers of GFR [28, 29], non-linearity of renal function reductions over time [30], declining in GFR for the first 3–4 months of therapy could be the reflections of reversible hemodynamics changes attributable to the therapeutic effects rather than true progressions of CKD [31, 32], GFR, and SCr analysis could be influenced by other several variables, and finally lack of confirmatory concordance evidence of the treatment effects of DSC and mortality.

In clinical practices, both clinical and surrogate end points have been used to evaluate DKD and CKD progression [33]. However, use of DSC as a surrogate end point to predict the DKD in patients with diabetes has not yet been studied in Ethiopia. Therefore, the main purpose of this five-year retrospective study was to determine the time when the SCr levels be doubled measured from their initial values SCr of patients with diabetes attending at the outpatient follow-up clinics of the University of Gondar Comprehensive Specialized Hospital (UoGCSH), Northwest Ethiopia.

## Materials and methods

### Study design and setting

A retrospective follow-up study was conducted from 2016 to 2020 on the medical records of patients with DM at UoGCSH, Northwest Ethiopian. Before the actual data collection, we ensured that the patients' medical records were documented between 2016 and 2020 and the data collection commencement date was started in January and completed in April 2021. The University hospital has been serving as a referral center for more than 7 million people annually, of which most of them are rural dwellers with poor socio-economic situations and about eight thousands of them were patients with DM [34]. In attempt to determine the time to which the serum creatinine level is doubled measured from the initial values of the patients with diabetes treated for diabetes and its complications with several medications, including antihypertensive medications, a hospital-based five-year retrospective follow-up study design was applied. A minimum of one year and a maximum of five years follow-up data were used to assess the time to be doubled the SCr levels. A one-year follow-up is used to rule out the potential reversible hemodynamics serum creatinine measurement changes attributable to the therapeutic reflective effects rather than the true progression levels. Further, screening once a year is recommended to assess the renal function status in all patients with diabetes comorbid with other complications [35].

### Study participants and sampling procedure

Adult patients aged 18 years and older who were diagnosed with DM were the study participants. To be included in the study, (1) patients with diabetes and greater than or equal to 18 years; (2) Had a minimum of 1-year follow-up (3) Had at least one record of SCr after a baseline within 365 days before study start. Whereas patients' medical records with a maximum of less than a 1-year follow-up data and patients who were on dialysis, had previous kidney transplant were excluded in this study. This longitudinal retrospective follow-up study was conducted on the medical documents of patients with diabetes attended in chronic medical ambulatory clinic of the hospital. Once the medical record of the identification number was

entered into the Microsoft excel 2013 and checked for repetition, the data were collected retro-spectively. The sample size was determined in compliance with a single population formula by considering the following assumptions; the response distribution, P = 0.5 (50%) was considered the rate of DSC levels in patients with diabetes. Using the freely available software Epi-info, we assumed 5% for two-tailed type-I error (Zα = 1.96); 80% power; two-sided 95% confidence level [26] and 0.35 hazard ratios of DSC level, and the Fleiss WC resulted 366 samples. Finally, 387 patients' medical records were included in the study after assuming potential missed and lost data. Then, using a simple random sampling method participant who met the inclusion criteria were included in the study using their medical identification numbers and the data was extracted, respectively, until adequate sample size was maintained.

## Outcome measurements

The time at which the first records in doubling of serum creatinine levels (measured in mg/dL) from the baseline values were used as cut-off points to have an outcome or developing doubling of serum creatinine (DSC) measures was amongst the composite significant kidney event end point indicators [16]. All the serum creatinine (SCr) values were measured from the medical files after a year of the first SCr levels were recorded as a baseline measure.

## Data collection procedures and quality control techniques

Clinical data were extracted using a semi-structured data collection tool. The data abstraction format was prepared after reviewing different related clinical literature on similar topics and some modifications were made considering the local clinical settings. Before the actual data extraction, pre-test was conducted on 20 medical records in the study area to ensure the completeness of the abstraction format. The pre-tested medical records were excluded from the final analyses. Then, an appropriate amendment was made. Data was extracted by experienced clinical nurses after they received training for three consecutive days about the data collection instruments and ethical aspects. The supervisor explicitly clarified the purpose of the study and data abstraction tool; and monitored the data collection closely. First, based on the lists, data (baseline SCr and other variables) were extracted from those medical archives, retrospectively until the first occurrence of the expected outcome documented with a maximum of 5 years follow-up. Recordings that were defaulted and lost during the retrospective follow-up were considered as censored.

## Data entry processing and statistical analyses

Data were entered in to EPI-Info version 7.0 and then transformed to the Statistical Package for Social Sciences (SPSS) version 26 for analysis. Descriptive statistics like frequency tables, graphs and cross tabulations were presented to characterize the study participants. Q-Q plot, histogram and Shapiro-Wilk test were used for the test of Normality. Chi-square ($X^2$) test was also performed to compare the baseline socio-demographics and clinical characteristics, co-morbidities, medications, and dosage strength differences between the comparative groups. Mann Whitney U tests and independent samples T-test for median and mean differences were employed, respectively. Times to DSC values and changes were predicted by using of the Kaplan-Meier procedure. The Log-rank test and cox-proportional hazard regression model were also used to test if there were significant changes in the SCr among different groups of predictor variables. $P < 0.05$ at 95% CI was statistically significant.

## Ethics considerations

First, the proposal of this study was ethically approved by the Ethical Review Committee (ERC) of the school of pharmacy and the Institutional Review Board (IRB) of the University of Gondar with reference number SOPS/043/2020. Consent to participants directly from patients was not applicable and instead it was waived by the IRB since the study was conducted on the medical record of patients retrospectively. Then, to proceed the data extraction, an official permission letter was obtained from the clinical director of University of Gondar Comprehensive Specialized Hospital (UoGCSH). The collected data was sufficiently anonymized, and patient identification could not be possible, and the information obtained from the study was not disclosed to the third party. Only coded numbers were used to identify the study participants.

# Results

## Socio-demographic and clinical characteristics of the participants

This study included 387 participants. The majority of the participants were females (54.5%) with a mean (±SD) age of 61.1±10.3 years. Of the included participants, T2DM patients (91%) and urban dwellers (71.6%) were over represented. Almost more than seventy percent of the samples had been stayed ≥ 5-years with DM. Likewise, significant numbers of the study patients were with uncontrolled plasma glucose levels (72.6%). All the sampled patients had been taking one or more antidiabetic agents and antihypertensive medications (**Table 1**).

**Table 1. Socio-demographic and baseline characteristics of patients with diabetic (N = 387).**

| Variables | Categories | Frequency (%) |
|---|---|---|
| Sex: | Male | 176 (45.5) |
| | Female | 211 (54.5) |
| Age (in years): | <60 | 180 (46.5) |
| | ≥60 | 207 (53.5) |
| Residence: | Rural | 110(28.4) |
| | Urban | 277(71.6) |
| Type of DM: | T1DM | 35(9.0) |
| | T2DM | 352(91) |
| Initial metabolic syndrome: | No | 334 (86.3) |
| | Yes | 53 (13.7) |
| Initial Macrovascular complications: | No | 352 (91) |
| | Yes | 35 (9) |
| Initial Microvascular complications: | No | 360 (93) |
| | Yes | 27 (7) |
| Initial renal events: | No | 376 (97.2) |
| | Yes | 11 (2.8) |
| DM duration (years): | <5 | 76 (19.6) |
| | ≥5 | 311 (80.4) |
| Initial FBS (mg/dl): | <130 | 106 (27.4) |
| | ≥130 | 281(72.6) |
| Glycemic control: | Good | 106 (27.4) |
| | Poor | 281(72.6) |
| Initial SBP (mmHg): | <140 | 155 (40.1) |
| | ≥140 | 231 (59.9) |

(*Continued*)

**Table 1.** (Continued)

| Variables | Categories | Frequency (%) |
|---|---|---|
| Initial DBP (mmHg): | <80 | 196 (50.6) |
| | ≥80 | 191 (49.4) |
| Initial DM agents: | Metformin | 71 (18.3) |
| | NPH | 92 (23.8) |
| | Glibenclamide | 10 (2.6) |
| | Metformin+glibinclamide | 136 (35.1) |
| | Metformin +NPH | 56 (14.5) |
| | Metformin+glibinclamide+NPH | 22 (5.7) |
| Initial ACEIs/ARBs type: | Enalapril | 380 (98.2) |
| | Candesartan or Losartan | 7 (1.8) |
| Initial other BP lowering agents: | Beta Blockers | 9 (2.3) |
| | Diuretics | 9 (2.3) |
| Initial lipid lowering: | Atorvastatin | 99 (25.6) |
| | Simvastatin | 53 (13.7) |
| Acetyl salicylic acid (ASA) | | 61 (15.8) |
| Amitriptyline | | 12 (3.1) |
| Asthma medications | | 6 (1.6) |
| Antimetabolites | | 6 (1.6) |
| [a]Other drugs | | 8 (2.1) |

[a] Other drugs includes (omeprazole, digoxin, PTU, KCl, ART, warfarin).

Throughout the follow-ups, smaller number of the participants (<10%) had developed a variety of cardiovascular, renal and metabolic complications. Either single or combined pharmacotherapeutic regimens were instituted for managing of these complications (**Table 2**).

**Mean serum creatinine measurements across different months of follow-up.** The trend graph illustrated below demonstrates the mean SCr difference detected between the baseline scores with the values recorded at different follow-up months. The subjects being studied had been followed for a mean (±SD) of 31.7±15.5 months (ranges: 12 to 60 months). At the beginning of the start of the study, the mean (±SD) level of the SCr was 0.87 (±0.23) (mg/dL), whereas it was 1.0(±0.37) mg/dL at the end of the follow-up. Finally, when the study ends, the overall mean difference (last minus baseline) was 0.12(±0.38) mg/dl. Hereby, the trend highlighted that it was increasing in the successive follow-up years and was documented as 0.94 (±0.31), 0.95 (±0.33), 0.94 (±0.25), 1 (±0.32) and 1.02 (±0.29) mg/dL 1-5years, respectively (**Fig 1**).

The One-sample T-test depicted that the mean level of the baseline SCr was 0.873(±0.23) with a 95% CI (0.85–0.896; p<0.001) and the mean SCr levels at the end of the follow-up was 0.995(±0.37) with a 95% CI (0.958–1.03; p<0.001).

**Incidence of doubling of the serum creatinine from the baseline among patients.** Of the total subjects took part in the study, just more than 10.3% of participants had developed the DSC level measured from the baseline. The estimated incidence rate of having DSC levels from the baselines in the patients with DM was about 22 per 1000 person-years (PYs).

## Kaplan–Meier hazard curve of the doubling of serum creatinine from the baseline

The Kaplan-Meier hazard curve had examined the time to doubling of the serum creatinine measured from the baseline. The mean survival time taken for doubling of the serum

**Table 2. Medical complications and medications types that the study subjects taking during the follow-up times.**

| Variables | | Frequency (%) |
|---|---|---|
| Complications: | ASCVDs | 17 (4.4) |
| | Microvascular | 15 (3.9) |
| | Renal complications | 9 (2.3) |
| | Metabolic syndrome | 20 (5.2) |
| DM medications: | Metformin | 58 (15) |
| | NPH | 105 (27.1) |
| | Metformin+glibinclamide | 83 (21.4) |
| | Metformin +NPH | 112 (28.9) |
| | Metformin+glibinclamide+NPH | 29 (7.5) |
| Lipid lowering agents: | Atorvastatin | 191 (49.4) |
| | Simvastatin | 154 (39.8) |
| | Lovastatin | 42 (10.9) |
| Other BP lowering agents: | Beta Blockers | 18 (4.7) |
| | Diuretics | 14 (3.6) |
| Acetyl salicylic acid | | 66 (17.1) |
| Amitriptyline | | 30 (7.8) |
| Antibiotics | | 7 (1.8) |
| Analgesics | | 6(1.6) |
| Antimetabolites | | 2 (0.5) |
| [a]Other drugs | | 4 (1) |

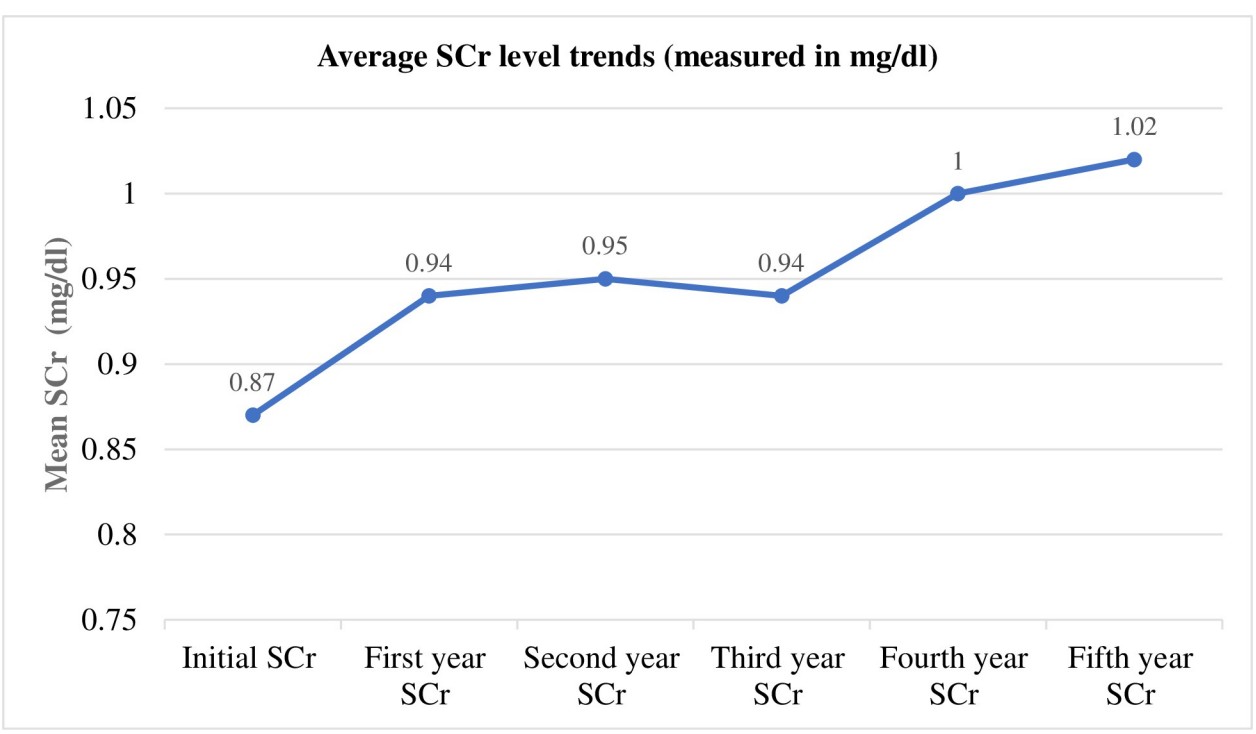

**Fig 1. The mean serum creatinine measurement trends (mg/dl) across different months of the follow-up in patients with diabetes mellitus.**

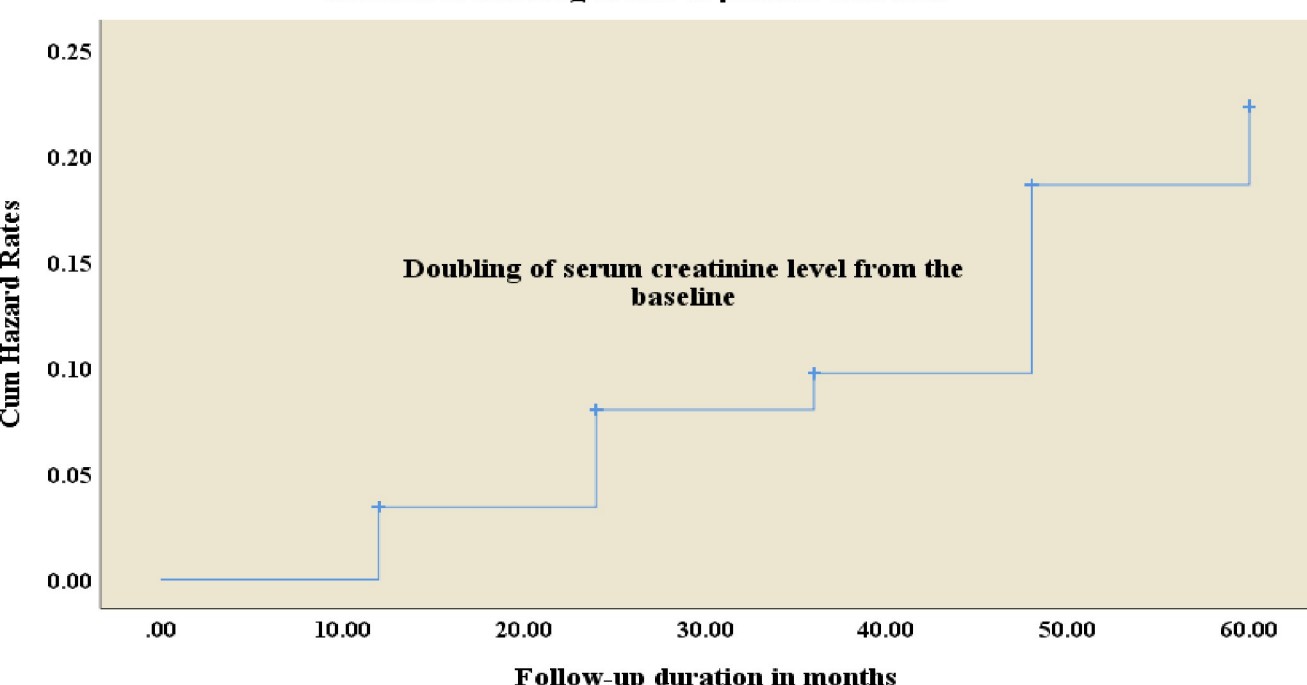

**Fig 2. Kaplan–Meier cumulative hazard rate curves depict the doubling of serum creatinine levels compared the baseline in patients with diabetics.**

creatinine levels from the baseline was 55.4 months (4.6 years) [Standard error (SE) 0.7] with a 95% CI (54.0 to 56.8) (**Fig 2**).

The univariate analysis highlighted that the study subjects with diastolic BP level was $\geq$ 80mmHg had lower likelihood to have risks of getting DSC by 58% [Crude hazard ration [36] (95%CI): 0.42 (0.21–0.85), p = 0.015]. However, this effect was not continued up on the multivariate analysis. On the other hand, patients initially treated with Glibenclamide, combination of Metformin and NPH, and triple combinations of Metformin, glibenclamide and NPH had about 14, 5.5 and 8 times more likely to have higher hazard of developing DSC level with [CHR (95%CI): 15.05 (2.75–82.28), p = 0.002]; [CHR (95%CI): 6.38 (1.4–29.12), p = 0.017]; and [CHR (95%CI): 8.02 (1.6–39.79), p = 0.011], respectively compared with patients who were on metformin alone. Nevertheless, this difference also had a lack of apparent associations in the multivariate analysis. The multivariate cox-regression analyses showed that the mean dose of NPH was significantly associated with the occurrences DSC. Hereby, individuals who had been treated with intermediate insulin (NPH) with dosage strength of $\geq$30 IU had about 3.3 times higher hazards in increasing for developing the doubling of the serum creatinine level from the baseline, with HR of 3.29 (95%CI), 1.28–8.44, P = 0.013 (**Table 3**).

## Discussion

This five-years retrospective follow-up study has gone some way towards enhancing our understanding on the importance of measuring the time to which the serum creatinine level is doubled that is computed from the baseline as a biomarker and predictor of diabetic kidney diseases among patients with DM in the resource-limited settings. In this study, we obtained comprehensive results proving that the serum creatinine scores of patients with diabetes could

**Table 3. Predictors and hazard ratios (95% CI) for doubling of serum creatinine in the five-year follow-up in patients with diabetes.**

| Variables | | Unadjusted Hazard Ratio (95%CI) | P-value | Adjusted Hazard Ratio (95%CI) | P-value |
|---|---|---|---|---|---|
| Age (in years): | <60 | 1 | | 1 | 0.623 |
| | ≥60 | 1.78 (0.92–3.45) | 0.088 | 1.23 (0.55–2.75) | |
| DM duration (years): | <5 | 1 | | | |
| | ≥5 | 27.01(0.57–1274.1) | 0.094 | | |
| HTN duration (years.): | <5 | 1 | | 1 | 0.810 |
| | ≥5 | 1.94 (0.81–4.64) | 0.137 | 1.12 (0.44–2.85) | |
| Initial DBP (mmHg): | <80 | 1 | | 1 | 0.12 |
| | ≥80 | 0.42 (0.21–0.85) | 0.015 | 0.513 (0.22–1.19) | |
| Initial DM agents: | Metformin | 1 | | | |
| | NPH | 3.89 (0.84–18.01) | | | |
| | Glibenclamide | 15.05 (2.75–82.28) | 0.002 | | |
| | Metformin+glibinclamide | 2.36 (0.51–10.92) | | | |
| | Metformin +NPH | 6.38 (1.4–29.12) | 0.017 | | |
| | Metformin+glibinclamide+NPH | 8.02 (1.6–39.79) | 0.011 | | |
| Initial renal events: | No | 1 | | 1 | 0.077 |
| | Yes | 3.01 (0.93–9.76) | 0.067 | 3.29 (0.879–12.299) | |
| Mean dose of NPH [23]: | <30 | 1 | 0.02 | 1 | 0.013* |
| | ≥30 | 2.35 (1.15–4.82) | | 3.29 (1.28–8.44) | |
| Lipid lowering agents: | Atorvastatin | 1 | | 1 | 0.051 |
| | Simvastatin | 0.737 (0.379–1.432) | 0.368 | 0.48 (0.214–1.09) | |
| | Lovastatin | 0.471 (0.141–1.57) | 0.22 | 0.125 (0.015–1.023) | |

* Indicates, variables at P < 0.05.

be doubled within less than five years. Further, DSC is an important surrogate end point indicator of renal function that could provide considerable insight and one of the acceptable alternatives to the clinical end points.

To demonstrate the composite renal end points, the patients with diabetes were followed for an average of 32 months (~2.67years); however, the mean time in which the serum creatinine was being doubled observed within about 55.4 months (~4.62 years). Broadly speaking, the DSC values are used as a biomarker of the renal composite end points and are hardly distinguishable from the subsequent clinical risks of ESKD and death. Further, the treatment effects of DSC are fairly similar to ESKD; although the DSC measurements have meaningfully higher number of events than death and ESKD in a relatively shorter period. Therefore, to achieve the required events, the number of required sample sizes and the follow-up time is significantly reduced compared to the clinical end point indicators such as death and ESKD [16]. On the other hand, DSC corresponds to 55% and 57% reduction in eGFR [16] if the MDRD and CKD-EPI equations are used [17, 37, 38], respectively. The present findings offer compelling evidence for clinicians and researchers to provide clinical decisions in a short period and with smaller sample sizes. This might also have important implications for solving challenges faced by the estimated glomerular filtration rates (eGFR), ESKD and death. Moreover, earlier studies endorsed to use the DSC as a biomarker for detecting and predicting the renal function status and providing greater clinical implications [21–23].

This study also provides further evidence for the magnitude of the observed mean serum creatinine differences of patients with diabetes. At the initial stage of the longitudinal follow-up, the mean level of the SCr was 0.87 (±0.23) (measured with mg/dL). However, it had been increasing in the succeeding follow-up years and the last mean SCr score was higher than the

previous values recorded during the follow-up time 1.0 (±0.37 mg/dL). Furthermore, at the end of the study, the mean difference from the baseline was 0.12mg/dl. This score is correlated favorably well with the previous studies [39, 40]. This observed difference has several implications for that group of patients who have a greater hazard of developing diabetic kidney diseases. In patients with diabetes, an observed substantial change in serum creatinine could be an indicator of decline in renal function, and the DSC has been a marker of renal composite end points for a long time in the studies of clinical nephrology [19, 41, 42]. DSC is also highly associated with the occurrence of several cardiovascular complications [36]. In this regard, studies to examine the incidence of DSC and the time of occurrence in patients with diabetes might be crucial to provide evidence-based interventions to prevent the worsening of the renal outcome of our patients. Particularly, in settings where there are several risky patients with diabetes are increasing and the renal function tests are not done routinely.

At the end of the follow-up, nearly more than 10% of the patients were found to have DSC levels from the baseline. The estimated incidence rate of developing the DSC measures from the baselines in the patients with diabetes was about 22 per 1000 person-years (PYs). These can be accounted for in part by inappropriate intervention and less intention might be given for monitoring. Besides, this study was conducted in developing country where there is a possibility of inadequate implementation of the therapeutic interventions. Therefore, with an appropriate management and monitoring practices, this condition could be reversed and a significant number of patients could be prevented from developing these events.

The current study also showed those patients with diabetes who were treated with more than 30 IU of an intermediate acting insulin regimen (NPH) had a higher threat of accelerating the development of doubling of the serum creatinine level, as computed from the initial scores. This could be therefore conceivably hypothesized that higher doses of insulin might result in another renal clearance burden for self-excretion and even it could further compromise the renal function of patients with diabetes. These patients already have an initial risk of renal deterioration because the kidney plays an important role in excreting insulin from the systemic circulation. The finding also suggests that patients with diabetes are at a high risk of renal decline might require insulin dosage adjustment on the basis of SCr scores and eGFR status [43, 44]. The results justified those patients with diabetes potentially result in renal problems over a period. Medications like higher dose of insulin might also contribute partly for the risk of increasing serum creatinine scores. Early monitoring and screening of renal function in patients with diabetes might be crucial so that poor renal outcomes could be prevented. Particularly in poor healthcare settings like Ethiopian where the renal functions tests are not measured routinely especially in high-risk patients with diabetes.

Generally, this study may implicate that the rapid rise in the prevalence and burden of diabetes mellitus in developing countries like Ethiopia is an urgent call for prompt secondary and tertiary prevention to minimize chronic complications, particularly renal events, which are related with the disease.

## Study strengths and limitations

The use of DSC as a measure of the surrogate end points of the composite renal outcomes since it correlates well with the subsequent risks of the clinical end points is the study strength. Further, in a relatively shorter period and smaller samples. the DSC possibly predict an outcome than death and ESKD. Hence, this study could have widened our clinical knowledge and have many implications for the clinical decision-making processes in the present clinical practices in the resource-limited healthcare settings. However, because of its retrospective nature of the study which could have potentially missed some theoretical variables; such as

proteinuria, albuminuria to creatinine ratio and there could be uncontrolled confounders. It was also single-centered study and there was incapability to generalize the finding. As a result, all the findings should be interpreted cautiously and prospective studies with the inclusion of all possible factors would be highly recommended. Though the DSC is commonly used, the validity of using this outcome as part of a composite endpoint is hampered by various factors such as the SCr levels might depend on the muscularity of the individual patients, the serum creatinine variations can reflect the hemodynamic changes in renal perfusion and not have a structural effect on renal function and the DSC could not perfectly reflect the composite clinical end points. As a result, all the findings should be interpreted cautiously.

## Conclusion

The evidence from this study suggests that a significant proportion of patients with diabetes developed DSC levels within approximately less than five years. Moreover, this study has confirmed that the mean level of the SCr was continuously and meaningfully increasing during the follow-up years computed from the baseline level. To preserve the renal function of patients with DM, regular and periodic monitoring and screening of renal functions would be important, particularly in the poor healthcare settings where the renal function tests are not routine.

## Supporting information

**S1 Data.**
(SAV)

## Acknowledgments

We would like to acknowledge the Clinical Pharmacy department, University of Gondar, college of medicine and health sciences, medical directorate for inspiring us to do this project; the authors also would thank the data collectors.

## Author Contributions

**Conceptualization:** Adeladlew Kassie Netere, Ashenafi Kibret Sendekie.

**Data curation:** Adeladlew Kassie Netere.

**Formal analysis:** Adeladlew Kassie Netere, Ashenafi Kibret Sendekie.

**Investigation:** Ashenafi Kibret Sendekie.

**Methodology:** Adeladlew Kassie Netere, Ashenafi Kibret Sendekie.

**Project administration:** Ashenafi Kibret Sendekie.

**Resources:** Adeladlew Kassie Netere.

**Software:** Adeladlew Kassie Netere, Ashenafi Kibret Sendekie.

**Supervision:** Adeladlew Kassie Netere.

**Validation:** Adeladlew Kassie Netere, Ashenafi Kibret Sendekie.

**Visualization:** Ashenafi Kibret Sendekie.

**Writing – original draft:** Adeladlew Kassie Netere, Ashenafi Kibret Sendekie.

**Writing – review & editing:** Adeladlew Kassie Netere, Ashenafi Kibret Sendekie.

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
