## [Decision Letter · Decision Letter 0]

18 Jul 2022

PONE-D-22-12838Time to Doubling of Serum Creatinine in Patients with Diabetes in Ethiopian University Hospital: Retrospective Follow-up StudyPLOS ONE

Dear Dr. Netere,

Thank you for submitting your manuscript to PLOS ONE. After careful consideration, we feel that it has merit but does not fully meet PLOS ONE’s publication criteria as it currently stands. Therefore, we invite you to submit a revised version of the manuscript that addresses the points raised during the review process.

We look forward to receiving your revised manuscript.

Kind regards,

Donovan Anthony McGrowder, PhD., MA., MSc

Academic Editor

PLOS ONE

Journal Requirements:

a) Did participants provide their written or verbal informed consent to participate in this study?

Additional Editor Comments:

Dear Dr. Netere,

 Your manuscript “Time to Doubling of Serum Creatinine in Patients with Diabetes in Ethiopian University Hospital: Retrospective Follow-up Study” has been assessed by our reviewers. They have raised a number of points which we believe would improve the manuscript and may allow a revised version to be published in PLOS ONE. Their reports, together with any other comments, are below.

 If you are able to fully address these points, we would encourage you to submit a revised manuscript to PLOS ONE.

Kind Regards,

Dr. Donovan McGrowder (Academic Editor)

Reviewers' comments:

Reviewer's Responses to Questions

**Comments to the Author**

1. Is the manuscript technically sound, and do the data support the conclusions?

Reviewer #1: Partly

Reviewer #2: No

2. Has the statistical analysis been performed appropriately and rigorously? 

Reviewer #1: Yes

Reviewer #2: I Don't Know

3. Have the authors made all data underlying the findings in their manuscript fully available?

Reviewer #1: No

Reviewer #2: No

4. Is the manuscript presented in an intelligible fashion and written in standard English?

Reviewer #1: No

Reviewer #2: No

5. Review Comments to the Author

Reviewer #1: General comments

The rapid rise in the prevalence and burden of diabetes mellitus in developing countries like Ethiopia is an urgent call for prompt secondary and tertiary prevention to delay or minimize chronic complications associated with the disease. The authors have used serum creatinine as an important biomarker and predictor of Diabetic Kidney Diseases among diabetic patients.

The authors have used a rigorous statistical analysis to respond to the study design and objectives.

However, the manuscript lacks line numbers and has many grammatical errors hence extensive English language editing is highly needed throughout the manuscript.

Specific comments:

Methods:

Study setting and design

Page 3: The sentence “In attempt to determine the time to be doubled the serum creatinine

levels of the patients with diabetes treated for dual antihypertensive medications, a hospital-based

five-year retrospective follow-up study design was applied.” Does this mean that diabetic patients were treated with dual hypertensive? Were they hypertensive as well? Or did you mean dual antidiabetics? It seems that your study eligibility criteria included only diabetic patients and not both diabetes and hypertension. Please clarify this matter.

Use Chronic Kidney Diseases if patients with both diabetes and hypertension were considered as eligibility criteria. Otherwise, use Diabetes Kidney diseases if the sample included only patients with diabetes.

Definition of terms

Page 4: The operational definition is not clear, too long and lacks reference. Also, the definition of macrovascular or cardiovascular disease is not necessary for this study.

Results:

In the field text result, the authors should present a few striking results and the rest can be found in the respective table(s).

References:

Some of the references are not written in the correct style as per the Journal guideline. For example, references # 3, 4, 16 & 32.

Reviewer #2: Netere et al. investigated chronological changes of serum Cr in CKD-DM patients. It may be useful for the local clinical practice. However, there are several significant issues.

1. BS control levels were not considered. Progression of CKD differs depending on individual BS control levels. The authors should analyze the data separately (e.g., different A1C levels). This should be applied to HTN control as well.

2. Urinalysis was not included. The authors should show those data.

3. Although novelty and significance should not be considered for this journal, the result is almost common sense.

---

## [Author Response · Author response to Decision Letter 0]

12 Aug 2022

Responses to the review’s comments

Dear PLoS ONE Academic Editor,

Thank you for giving us the opportunity to submit a revised draft of the manuscript and we would also like to thank your constructive and fruitful comments and suggestions on our paper (Manuscript ID: PONE-D-22-12838). We are very concerned and combined all the suggested comments provided, which we believe that strengthened our paper and we hope this render our paper to be considered for publication in your reputed journal. We appreciate the time and effort that you and the reviewers dedicated to providing feedback on our manuscript and are grateful for the insightful comments on and valuable improvements to our paper.

We authors would like to let you know that all comments and concerns raised by both academic editors and Reviewers are fully addressed using a MS-word indicated with track changes with a point-by-point response letter. Moreover, we did our best changes and corrections on this revised manuscript. All the changes and corrections are replied and indicated for a point-by-point response letter to the reviewers’ comments and concerns. All page numbers refer to the revised manuscript file with tracked changes.

Comments from the editor:

1#.... Journal requirements:

 Please ensure that your manuscript meets PLOS ONE's style requirements, including those for file naming

Author reply: Thank you for your recommendations to assure adherence to the Manuscript Template requirements of the journal. Considering to your recommendation, we have adjusted and ensured it accordingly. 

2#.... Please amend your current ethics statement to address the following concerns:

a) Did participants provide their written or verbal informed consent to participate in this study?

Author reply: Thank you very much for your request to clear regarding informed consent to participants. Written or verbal informed consent was not applicable in the study, since the study did not involve the patients directly because it was done on the patients’ medical records retrospectively. Instead, the ethical approval was waived by the IRB of University of Gondar. Considering your suggestions and concerns we revised and make clear it in the ethical consideration section of the manuscript. 

Author reply: Yes, the IRB approved the proposal and waived the consent procedure to conduct the study on medical records of the patients retrospectively. 

3#...... In your Data Availability statement, you have not specified where the minimal data set underlying the results described in your manuscript can be found. PLOS defines a study's minimal data set as the underlying data used to reach the conclusions drawn in the manuscript and any additional data required to replicate the reported study findings in their entirety.

Author reply: Thank you for your comments and suggestions to provide the minimal data set and revise the Data availability statement. Taking your suggestions positive, we had revised the Data availability statement and provided the minimal data set used to reach the conclusions drawn in the manuscript. 

4#.... We note that you have stated that you will provide repository information for your data at acceptance. Should your manuscript be accepted for publication, we will hold it until you provide the relevant accession numbers or DOIs necessary to access your data. If you wish to make changes to your Data Availability statement, please describe these changes in your cover letter and we will update your Data Availability statement to reflect the information you provide.

Author reply: Thank you for your suggestions regarding repository information and Data availability statement. By considering your comments and suggestion, we had provided the updated statement in the cover later and Data availability statement section. 

5#...... Please include your full ethics statement in the ‘Methods’ section of your manuscript file. In your statement, please include the full name of the IRB or ethics committee who approved or waived your study, as well as whether or not you obtained informed written or verbal consent. If consent was waived for your study, please include this information in your statement as well.

Author reply: Thank you very much updating the position of ethics statement in the method section and clear the ethical approval procedures. Based on your recommendations we updated the location to be at the method section and we have also cleared that the IRB of the University of Gondar and ERC of the school of pharmacy were approved the proposal and waived the consent. 

Response to Reviewers’ comments:

Reviewer #1

1#... General comments

The rapid rise in the prevalence and burden of diabetes mellitus in developing countries like Ethiopia is an urgent call for prompt secondary and tertiary prevention to delay or minimize chronic complications associated with the disease. The authors have used serum creatinine as an important biomarker and predictor of Diabetic Kidney Diseases among diabetic patients.

The authors have used a rigorous statistical analysis to respond to the study design and objectives. However, the manuscript lacks line numbers and has many grammatical errors hence extensive English language editing is highly needed throughout the manuscript.

Author response: We authors are very thankful for your deep concerns and suggestions. We, therefore, accepted the recommendations and made extensive English language editing and corrections for the whole manuscript that indicated with track changes. The manuscript has now prepared with line numbers and all the raised issues’ responses are indicated with those line numbers. 

Specific comments:

Page 3: The sentence “In attempt to determine the time to be doubled the serum creatinine levels of the patients with diabetes treated for dual antihypertensive medications, a hospital-based five-year retrospective follow-up study design was applied.” Does this mean that diabetic patients were treated with dual hypertensive? Were they hypertensive as well? Or did you mean dual antidiabetics? It seems that your study eligibility criteria included only diabetic patients and not both diabetes and hypertension. Please clarify this matter. Use Chronic Kidney Diseases if patients with both diabetes and hypertension were considered as eligibility criteria. Otherwise, use Diabetes Kidney diseases if the sample included only patients with diabetes.

2#... Methods: Study setting and design, Page-4; Lines 17-20, in tracked manuscript 

Author reply: Thank you very much for your request for clarification on the patient groups and medications they were taking and general eligibility criteria. 

Participants were patients with diabetes mellitus who were on follow-up. Unfortunately, majority of the patients were hypertensive in addition to diabetes, and they were taking several medications, as explained in the tabulation sections, including multiple anti-hypertensive agents for either blood pressure control or other cardiovascular conditions management or prevention. To be included in this study, the participants need to be patients treated for diabetes, though, most of those included subjects were also treated for hypertensive and were taking antihypertensives. Therefore, being hypertensive was not a mandatory to include subjects in this study and was not as an inclusive criterion. By considering your comments and suggestion, we had revised and cleared the participants and eligibility criteria. 

3#... Definition of terms

Page 4: The operational definition is not clear, too long and lacks reference. Also, the definition of macrovascular or cardiovascular disease is not necessary for this study.

3##---Methods section: Definition of terms sub-section, Page 5, lines 26-29 and page 6, lines 1-3, in tracked manuscript

Author reply: We authors are very kind to correct your concerns and suggestions, and we revised and amended it accordingly. By reminding you recommendations, we now believe that the Definitions of terms and Operational definitions are not important, and we decided to remove it from manuscript. The outcome definition we initially thought to be operationalized is already written in the method section, Outcome measurement sub-section page-5; lines 19-25. 

 #4.... Results:

In the field text result, the authors should present a few striking results and the rest can be found in the respective table(s). 

4##---Result section: Page 8-11, in tracked manuscript

Author reply: We really appreciate your valuable recommendations and suggestions on this section. By taking your suggestions and recommendations, we revised and rewrite the whole manuscript presentation and the field text presentations are reduced indeed. 

#5… References:

Some of the references are not written in the correct style as per the Journal guideline. For example, references # 3, 4, 16 & 32.

Author reply: Thank you for the valued comments to revise the references. Reminding your comments as positive and constructive, we made changes and corrections on the whole references accordingly.

Reviewer #2

Netere et al. investigated chronological changes of serum Cr in CKD-DM patients. It may be useful for the local clinical practice. However, there are several significant issues.

Comments:

#1… BS control levels were not considered. Progression of CKD differs depending on individual BS control levels. The authors should analyze the data separately (e.g., different A1C levels). This should be applied to HTN control as well.

Author reply: We authors are very grateful for the concerns and recommendations you raised on the BS for diabetes and BP for HTN. It is obvious that the current study did not rigorously analyses the BS and BP control levels as a potential predictor of DSC and DKD beyond using the baseline/initial BS and BP records, explained in Table-1 on the socio-demographic and baseline characteristics. Nonetheless, we understand that the BS and BP levels could partly determine the progression of CKD. On the other hand, our primary objective in this study was to assess the mean SCr levels and to determine the mean time to which the SCr level was being doubled as a biomarkers and predictors of diabetic kidney disease in patients with diabetic mellitus. Therefore, assessing BS for diabetes and BP for HTN could be secondary outcomes. Meanwhile, we tried to analyze the baseline/initial recorded data of the BS and BP control levels as a potential predictor of DSC and DKD. We further tried to determine whether those BS and BP control levels are potential factors for our outcomes that is the DCS time. By considering this, the initial FBS and BP (Table-1) were not fitted for further multivariate cox-regression model and the initial DB level was not significantly associated the outcome (Table-3). Furthermore, in our study we used FBS levels rather than HbA1c records, since HbA1c records was not commonly available in the study setting, and FBS level was used to determine the glycemic control. 

Moreover, we believe that to analyze the blood glucose and HTN control data separately (e.g., different A1C levels), might preclude us to do. 

1. The American diabetic association, 2019, recommends doing renal function test at least in a year to exclude the potential reversible hemodynamics SCr measurement changes attributable to the therapeutic reflective effects rather than the true progressions levels. Further, screening once a year is recommended to assess the renal function status in all patients with diabetes comorbid with other complication. So that determining the time required to the SCr level doubles could be the good indicator of DKD. 

2. On the other hand, BS and BP levels should be measured at least within three months interval to approximately estimate the potential prediction power to DKD and associations to DSC. Therefore, we thought that analysing these data separately with other study method and follow-up design might have good prediction power for the determinant factors to the DSC and DKD. 

#2…Urinalysis was not included. The authors should show those data.

Author reply: This is our cordial appreciation for your deep insight and important inputs on our paper and we shared your concerns and issues you raised. But as we try to mention in the study strengths and limitations section the study might potentially miss some important variables. This is because, the study design is a retrospective nature and we used the patients’ medical records that we could not find the Urinalysis consistently throughout the follow-up periods and we forced to not included it. Consequently, we tried to explain this in the study strengths and limitations section which is indicated with track changes page-15, lines 22-27 (in track changed manuscript). And it is true it might hamper the generalizability as a true predictor to DKD. So, we recommended that the findings would be interpreted in cautiously, and we the prospective study design might be address this shortcoming. 

3#…Although novelty and significance should not be considered for this journal, the result is almost common sense.

Author reply: We are very grateful for the concerns and your frustrations raised on the significance of results. Despite we could share your concerns, we believe that the findings of our study have several unique characteristics and clinical decision power especially in poor health care settings where the other surrogate (eGFR, CrCl) and clinical (dialysis, transplant and death) composite renal end points are not regularly applicable. Moreover, the results in this study could be used in the real-world clinical practices and the data is very crucial for clinical decision makers in such settings. Further, though it is a single centered study, it has still important to make changes in the resource limited health care settings, particularly in Ethiopia where there has not been investigated and published yet. So that, the current study might have relevant implications on some points towards adjusting the management and prevention strategies in those patient groups. Generally speaking, we believe that the DSC has a potential biomarker and indicator for DKD could have clinical relevance in Ethiopia where the rapid rise in the prevalence and burden of DM is increasing and need an urgent call for prompt secondary and tertiary prevention to minimize chronic complications particularly renal events which are related with the diabetes.

---

## [Editor Report · Decision Letter 1]

30 Aug 2022

Time to doubling of serum creatinine in patients with diabetes in Ethiopian University Hospital: Retrospective follow-up study

PONE-D-22-12838R1

Dear Dr. Netere,

We’re pleased to inform you that your manuscript has been judged scientifically suitable for publication and will be formally accepted for publication once it meets all outstanding technical requirements.

Kind regards,

Donovan Anthony McGrowder, PhD., MA., MSc

Academic Editor

PLOS ONE

Additional Editor Comments:

Dear Dr. Netere, 

The manuscript was revised in accordance with the reviewers’ comments and is provisionally accepted pending final checks for formatting and technical requirements.

Regards,

Dr. Donovan McGrowder (Academic Editor)<o:p></o:p>

---

## [Editor Report · Acceptance letter]

2 Sep 2022

PONE-D-22-12838R1 

Time to doubling of serum creatinine in patients with diabetes in Ethiopian University Hospital: Retrospective follow-up study 

Dear Dr. Netere:

I'm pleased to inform you that your manuscript has been deemed suitable for publication in PLOS ONE. Congratulations! Your manuscript is now with our production department. 

Kind regards, 

on behalf of

Dr. Donovan Anthony McGrowder 

Academic Editor

PLOS ONE